# Identities in Troubled Times: Minoritized Youth in Hong Kong's "Summer of Protest"

**Kerry J. Kennedy** [1,2,*], **Jan Christian Gube** [1] and **Miron Kumar Bhowmik** [3]

1    Department of Curriculum and Instruction, The Education University of Hong Kong, Hong Kong 999077, China; cgube@eduhk.hk
2    Department of Education and Curriculum Studies, University of Johannesburg, Johannesburg 2092, South Africa
3    Department of Education Policy and Leadership, The Education University of Hong Kong, Hong Kong 999077, China; mbhowmik@eduhk.hk
*    Correspondence: kerryk@eduhk.hk

**Abstract:** Discursive experiences can contribute to shaping lives and their identities. For minoritized youth in Hong Kong, the 2019 protest movement provided many such experiences, although very little has been heard about them. Instead, reporting has focused on the experiences of the dominant Chinese population. This paper aims to highlight the voices of minoritized youth in relation to the social movement that dominated Hong Kong in the second half of 2019. It is well recognized that identity is not fixed and that there are more likely multiple identities that transition from one to the other. Yet little is known about the influences on identity formation and the processes that underlie them. This was the issue addressed here. The paper draws on Lacan's theory of identity in examining interviews involving minoritized youth and their engagement in Hong Kong's 2019 protest movement. It shows how individual responses to the movement differed, how the movement challenged identities, and how these challenges were resolved.

**Keywords:** minorities; identity; values; Hong Kong; protest movement

## 1. Introduction

Hong Kong is a Chinese city located on the southern tip of the country and a cosmopolitan metropolis in the heart of Asia. Yet the history of the city is steeped in a colonial past that reflects not only British power and hegemony but also the trappings of the Empire, which included the import of Indian police officers, Pakistani traders, and Nepalese Gurkhas. Later generations of these originally imported colonial families remain in Hong Kong joined by modern-day residents who continue to make their way from these and other countries. They are joined by large numbers of Filipina and Indonesian domestic helpers who serve the city's middle classes with little acknowledgment of their contribution to the city's development. There is also a lingering population of Westerners from places as far apart as Europe, Australia, and Canada, as well as small groups from Thailand, Korea, and Japan. Together, this current multicultural population represents around 8% of the city's population and is growing [1].

Reference to a "multicultural population" in the previous paragraph reflects the reality of Hong Kong's population statistics but it is also a sharp reminder that Hong Kong does not have a policy on multiculturalism [2]. Such a policy has not been a priority for successive governments, although there is an Equal Opportunities Commission and a Racial Discrimination Ordinance. The latter reflects an attempt to inject a note of fairness into an otherwise unequal society, but it in no way recognizes the complexities of a multicultural society. Yet minoritized youth, as part of the broader multicultural society, face these complexities every day and in so doing seek to understand who they are in a world of conflicting values, old and new.

## 2. Purposes

The remainder of this paper will focus on a series of events in Hong Kong's recent history that we refer to here as the 2019 protest movement [3–5]. It seeks to portray how the identities of a sample of Hong Kong's minoritized youth were influenced by this environment and what this means for understanding processes of identity construction in this context.

The following issues will be addressed:

- Hong Kong's minoritized population—from subject to heritage citizen.
- Identity formation—which theory?
- Methods used in the research.
- Cases of minoritized youth and the 2019 protests.
- Discussion of the results.

## 3. Hong Kong's Minoritized Population—From Subject to Heritage Citizen

One way to view Hong Kong's minoritized population is through the official statistics. They show that 8% of the Hong Kong population self-reported as ethnic minorities in a society dominated by either native-born or immigrant Chinese. This represented an increase of just over 70% from 2006 [1]. Further census figures explained that "80% of all ethnic minorities in Hong Kong were Asians . . . the majority being Filipinos and Indonesians, constituting 31.5% and 26.2% of the population of ethnic minorities respectively. This was followed by South Asians, who represented 14.5%. On the other hand, 10.0% of the ethnic minorities in Hong Kong were Whites" (p. 2). Yet these figures, so often used in academic work about the city's ethnic minorities, miss the broader narrative that constructs the lived experience of these residents.

The term "resident" is used here deliberately. The Chinese government incorporated the status of Hong Kong Permanent Resident (HKPR) into Hong Kong's Basic Law as a part of the establishment of the Hong Kong Special Administrative Region (SAR) [6]. In large part, this was because the Chinese Nationality Law did not make any citizenship provision for those other than Great Britain's former Chinese subjects. The only option for the colony's Indian, Pakistani, and Nepalese was either the British National (Overseas) passport, which enabled travel but not residence in Great Britain [7] (Dummett, 2006), or a reversion to their heritage citizenship. These administrative arrangements, put in place to facilitate Hong Kong's return to China in 1997, have operated ever since. New ethnic minority arrivals, as well as those long-term "British subjects turned residents", all with heritage citizenships, are regarded as "residents" rather than citizens. They exist in the midst of a dominant Chinese citizenry, its values, its politics its educational provision, and its prejudices.

There is one final lens to turn on Hong Kong's minorities. A report from the city's Legislative Council (2017) summarized the context for many ethnic minorities [8]:

> By and large, three ethnic groups from South Asia (i.e., Pakistani, Indian and Nepalese) are more susceptible to poverty risks in Hong Kong, partly due to fewer working members and more dependent children in the households. Their educational attainment and engagement in lower-skill jobs also affected their monthly household income (p. 11).

In 2014, the average poverty rate across a sample of Hong Kong households was 19.6%; however, in Pakistani households it was 73.4% and in Indian households it was 34.3% (p. 8). Problems identified for these groups were language, employment, and support services. This highlights that the lives of ethnic minorities are context-bound to the extent that using the essentialist category, "ethnic minority", masks the differences both with and between multiple minoritized groups and individuals. This has significant implications for the ways identities are formed: shaped by diverse rather than common experiences leading to even greater unpredictability in an uncertain world.

#### 4. Hong Kong's "Social Unrest"—Identity Construction in Troubled Times

Referred to locally as "social unrest", the 2019 protest movement in Hong Kong was, in reality, a political crisis that spiraled out of control [3–5]. It was a complex situation that is not easy to condense. What follows is an attempt, first, to analyze the main issues and then link them to issues of identity. As we pointed out earlier, the way the movement was perceived depended on how it was politically constructed. For the protestors, the issue was to further what they saw as the democratic development of Hong Kong to the point of independence from China. For the pro-Beijing supporters, the issue was one of separatism and a challenge to Chinese sovereignty. The bulk of the reporting on the protests has focused on Chinese participants since they made up the very large majority of the protestors. It has been pointed out [9] that minoritized groups have been largely invisible in the local media, and when they are reported it is usually in negative terms. We have sought to correct this deficit by exploring the engagement of Hong Kong's minoritized groups with the protest movement. For this paper, we seek to understand how such engagement influenced identity and what this meant for those involved.

A clue was provided about the difficulty of such an assessment [5] when the local media portrayed the engagement of minoritized individuals and groups. There were multiple and conflicting portrayals. Both the English and Chinese pro-democracy papers highlighted minoritized engagement as "heroic" support for the movement, whereas the pro-China press tended to describe the same actions as those of "villains" intent on undermining the city. There were other media reports that on a spectrum came somewhere in between the extremes of the strong pro-democracy and pro-China newspapers. In general, the media reports on both sides indicated that the protest movement did influence minoritized groups. Yet more nuanced investigation is needed to understand the impact on identity, which is the purpose of this paper.

#### 5. Identity Formation—Which Theory?

The well-known sociologist, Stuart Hall, summed up what has become a common way of conceptualizing "identity" in postmodern times [10]:

> The concept of identity . . . is... not... essentialist, but a strategic and positional one. (It) does not signal that stable core of the self, unfolding from beginning to end through all the vicissitudes of history without change; the bit of the self which remains always-already 'the same', identical to itself across time. . .. identities are never unified and, in late modern times, increasingly fragmented and fractured; never singular but multiply constructed across different, often intersecting and antagonistic, discourses, practices and positions (pp. 3–4).

This statement is explicit enough in articulating the move away from identity as a core and stable part of being human to what is a constantly changing "self" depending on time, context, and circumstances. Yet the above conceptualization does not point to the processes by which construction and deconstruction take place—what might be called the underlying mechanisms.

One approach to understanding such processes is that of Lacan, who talked about identity from a psychoanalytic perspective. Issues of identity were not the main focus of his work [11], but considering the malleability of identity, his references to it are insightful. Central to Lacan's thinking about identity was an awareness of "other" (meaning "awareness of self as a separate being" [12]) and the interaction that takes place between individuals and their "other"; but also, and more importantly, there is the OTHER ("the 'great other' in whose gaze the subject gains identity", p. 10). This "great other" (OTHER) could, on the one hand, be parents and friends and on the other, oppressive structures in society that lead subjects to doubt their "other". The gaze of the "OTHER" generates identity (or identities) and, according to Lacan, the great "OTHER" always represses the personal "other".

Power is never far from this process: identities change as subjects experience multiple gazes. The power of the "OTHER" can be understood considering Levi Bryant's comment [13] quoted by Hewitson (2013), that "the Lacanian subject is quite literally a void or emptiness. It's a sort of empty point, a mobile empty space...". Identities fill the space, sometimes consciously and sometimes unconsciously.

Gingrich (2004) suggested [12] that Lacan's position, and others like it, might be seen as a kind of "psychoanalytical fatalism" (p. 11). This raises an important issue about the agency of subjects, an issue also raised by Stuart Hall [10]. His view of identity, while not in the psychoanalytic tradition, is not inconsistent with it:

> Identities are thus points of temporary attachment to the subject positions that discursive practices construct for us (p. 6).

Hall did not wish to re-create, in Foucault's terms, the "knowing subject" (p. 11), but rather to focus on discursive practices and how these may impact directly on identity/identities. In this sense, subjects' identities may not be constructed by the "OTHER" alone, but also by the discursive practices in which they engage. This integrated view of identity formation is shown in Figure 1.

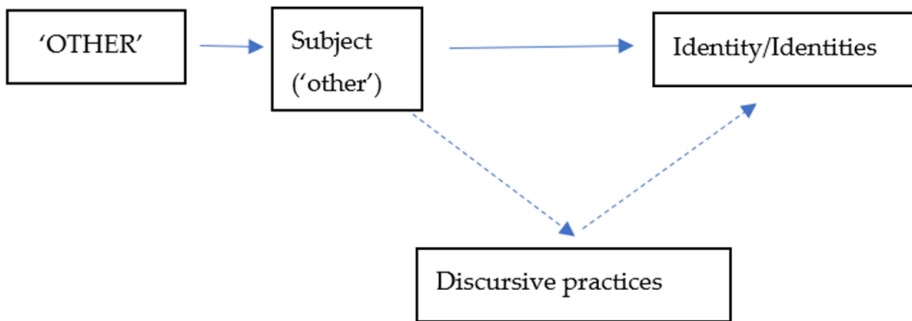

**Figure 1.** An integrated view of identity formation.

In this integrated model, the role of Lacan's "OTHER", as either a conscious or unconscious mechanism, is acknowledged; but so too is the agency of the subject engaging in, contributing to, and influencing discursive practices. Identities emerge from these multiple interactions. Such identities are both personal and social, an ever-present fluidity in the lives of subjects.

## 6. Methods Used in This Research

### 6.1. Our Sample

The sample used in the present study was part of a larger project that examined the attitudes of Hong Kong minorities to the 2019 protest movement [14]. In the larger project, purposive sampling was used to recruit 25 interviewees, including 13 community leaders and 12 individuals, with multiple ethnicities. Given the requirements of the present project to focus on youth, four interviewees in the larger project were selected for inclusion in the present project. The one exception was a 28-year-old Filipina since, as mentioned earlier, her ethnic group is the largest in Hong Kong. Details of this sub-sample are provided in Table 1.

### 6.2. Our Interviews

To conform with ethical requirements, participants were asked to sign a consent form prior to the interview, and they were provided with an information sheet about the research project. An interview protocol was developed and endorsed by the Research Ethics Committee at the authors' university. The interviews were semi-structured in nature and allowed participants to respond freely and for the interviewer to ask follow-up questions. Spontaneous follow-up probes can be helpful in seeking clarification and asking for more details from interviewees. Interviews were recorded and transcribed.

**Table 1.** Demographic characteristics of interviewees.

| Interviewee (Pseudonym) | Age | Sex | Ethnicity | Birthplace/Citizenship/ Duration of Residence in HK | Occupation |
| --- | --- | --- | --- | --- | --- |
| Malolos | 28 | F | Filipino | Hong Kong/Filipino/28 | Law firm marketer |
| Mizan | 20 | M | Indian | India/Indian/20 | Student |
| Amrita | 23 | F | Indian | India/HKSAR/23 | Teacher |
| Aqidah | 23 | F | Pakistani | Pakistan/Pakistani/17 | Student |

In line with the theoretical framing of the research, interviewers were not searching for commonalities or assuming that there was an essentialized "ethnic minority" identity. The focus was on each individual, the ways they constructed their identities, the factors that seemed to influence them, and how they used these identities to negotiate everyday living in Hong Kong.

*6.3. Data Analysis*

Interview transcripts were carefully read and themes were identified based on the categories identified in the interview protocol. The main points were then grouped into different themes related to the way interviewees described not just their involvement or non-involvement in protests but how such engagement influenced their identities. Each transcript was treated as an individual experience and as reported below, presented as an individual case. No attempt was made to compare and contrast individuals. This was in line with the theoretical framing of the project.

**7. Cases of Minoritized Youth in Hong Kong's 2019 Protest Movement**

*7.1. Amitra*

Amitra is a student of Indian heritage who was 23 years old at the time of the interview. She came to Hong Kong when she was two months old. She was a permanent resident (PR) of Hong Kong and a citizen of India. Like many in the city, she was initially supportive of the protest movement, becoming disillusioned on account of the violence that emerged. When asked what the protests meant for her personally and for her identity she replied:

> I think it did give me an identity, and not the form that you would expect. I don't think I'm a 'Hongkonger' that the protesters or the government shapes... I associate with a Hong Kong that is very open, welcoming, safe and a land full of opportunities. That is the Hong Kong I know. And that's the Hong Kong I've seen myself in all this time...So that is my identity. I've realized... why I like Hong Kong so much more than...I like India.

In other words, the protests made Hong Kong unrecognizable for Amitra. She stressed this point by comparing it with India:

> That is not what I've come to know of Hong Kong. I'm Indian and I know how corrupt India is, but Hong Kong is nothing like that... and seeing what I would see in Indian movies being portrayed in Hong Kong...was very much shocking. And I still remember the effect it had on me when I saw the images on the Instagram page.

In theoretical terms, Amitra was first influenced by the protests as "OTHER", but her discursive experiences, which were largely virtual, led her to reject them. They interrupted her own sense of being a Hongkonger and the values that she prized more than those being demonstrated by both the police and the protesters. She pushed back against the "OTHER", creating an identity shaped instead by her own values or in Lacan's terms by the "other".

### 7.2. Aqidah

Aqidah is a university student who came to Hong Kong with her family from Pakistan when she was very young. She was 23 years old at the time of the interview. She identified several problems with both the protesters and the police and their actions throughout the protests. Overall, however, she was supportive of the protesters' objectives although her own experiences at different times were not positive. Eventually, however, she stopped engaging in anything to do with the protests saying that such engagement affected her mental health. When asked about the impact of the protests on her identity she replied:

> It kind of did. . . I'm now confused. . . if I want to be part of this nation. . .Because after all, we are Chinese, right? We belong to China, no matter how much we deny it. . . eventually we will go back to China. . .but I don't want to go back. . .and I am confused. . .Like if I want to be part of this nation that that does not give its people the freedom of speech and it doesn't treat its people fairly. . . so yeah, I'm confused. If I want to be part of the nation or not in the future. This is how it affected my identity. Before I felt a part of it . . . living here and being part it...being a homeowner, but now I don't think so. Hong Kong is losing its freedoms one by one.

Aqidah's disillusion, which is so clear here, is based on her assessment that what she has valued in the past about Hong Kong is disappearing. Perhaps more importantly she fears what will replace it—she loves Hong Kong but fears China and what it might hold for her. She looks into the future and sees what is coming and she does not like it. Like Amitra, the protests have challenged Aqidah's conception of what life in Hong Kong has to offer. Yet she does not give up on the importance of the protests although she disagrees with the violence. As she said she is "confused"—meaning her identity is confused:

> Sometimes there is hope that everything will be normal. But then when you think that China is such a powerful country, and it has so much economic power and how it can manipulate its power. . . it's not clear...But right now, for right now. I want to graduate as a teacher because I have many goals as an ethnic minority.

Aqidah's aspiration to become a teacher to support her community overrides her confusion. It becomes a greater goal than anything to do with the protests or her fear that Chinese sovereignty will extinguish Hong Kong's freedoms. Her personal identity (based on her "other") was stronger than the "OTHER" in the form of the protests. It is as though her "other" resolved her confusion allowing her identity as a teacher to reemerge in her consciousness.

### 7.3. Mizan

Mizan is a university student of Indian heritage who was 23 years old at the time of the interview. His interview sounded, and the transcript reads, like a newspaper article. He was well versed in the protest movement and had participated indirectly, particularly through Telegram, the most popular messaging app used by the protesters. He acknowledged during the interview that "Hongkonger" identity was a driver for the protests. Yet he did not identify in this way himself. Reading and re-reading the transcripts, Mizan emerges as an analyst of the protest movement rather than an advocate, almost disengaged despite his sympathy. Towards the end of the interview, however, he highlighted the issue of his identity:

> In the middle of this for me. . . the identity of being a Muslim came up. Being a Muslim, I think, is really international in the sense that it gives you an international identity. So, wherever I go, I'm bound by Muslims everywhere. India, Pakistan, even here in Hong Kong, you go to China. . .So this is an international identity. . .. At times when I'm battling with these political convictions, like whether I should do this or choose that. . .well, I kind of look at it from an Islamic perspective . . .Islam literally has a totally different set of laws. A totally

different set of political. . .convictions which I feel are convenient. So basically, I'll weight both. . .situations from India and from in Hong Kong. I'll look at the commonalities between them, and basically. . .decide on from an Islamic perspective. Because. . . my religion does not meet the need of Western democratic ideals, because there are two totally different sets of rules and it's an entire worldview, in the sense that it actually tells you where to stand politically and how to. . .solve political issues as in like your convictions and, you know, in terms of rules.

Thus, despite Mizan's discursive experiences with the protest movement reflecting his sympathies for the protesters and seeing him use the Telegram app to lend physical support, his identity is not shaped by these experiences. In this sense, the protests do not act as "OTHER" for him. He is guided by the "OTHER" of religion and recognizes its role in his life. His own "other", or sense of personal values, enables him to participate virtually as a support for the protesters, but this kind of engagement does not appear to affect his identity. He has a larger set of principles that construct his identity, and it seems he is able to make decisions at multiple levels, but at no time do the decisions he makes override his "OTHER" of religion.

### 7.4. Malolos

Malolos is a young Filipina born in Hong Kong with her citizenship in the Philippines. While she had finished her A-levels and had been offered a place in a university in the United Kingdom, she did not take it up. She worked in the marketing area for a law firm. In response to the interviewer's question: "Do you mind telling us a little about your background?", Malolos immediately asserted that her identity was "Hong Kong first and Filipina second". She was the only interviewee who started the interview in this way, suggesting that being a "Hongkonger" was somehow foremost in her mind. Her subsequent interview showed why this was the case.

When the Interviewer asked her about her attitude toward the protests her response was unequivocal:

I work in Admiralty [an inner-city area] and I saw them storm Legco [the Legislative Council]. I was rooting for them. I was totally in favour of abolishing the Extradition Bill [by which Hong Kong people could be extradited to Mainland China if they were charged with breaking Mainland law] and I totally understand why it snowballed from an Anti-Extradition Bill protest into this whole movement against like police brutality and clashes. . .. yeah, very supportive.

Throughout the interview, Malalos demonstrated her commitment to the causes of the protesters indicating that on occasion she had joined peaceful protests and donated money to the cause. She explained her reasons:

I'm born and raised here, and I feel it's my responsibility to my community to make my voice heard.

When asked by the Interviewer, "What do you think the protests mean for you personally" she replied:

For me, it really is a question of what defines...Hong Kong or the ideological nature of what defines a Hong Kong. . . it's not about your skin colour or where you're from. It's about how you support the city and how you're fighting for it.

This linked to an earlier comment showing her sense of belonging to the city, even though the city's residents did not always recognize it:

I had responsibility to make this place better and to give my time to it, but also, I realized that this was a way to share something with the local community because I felt so ostracized prior to this.

For Malolos, the protests are "OTHER," reminding her of her Hongkonger identity and her sense of belonging to the city. It leads her to engage in discursive experiences

consistent with her "OTHER". She dismisses the violent aspects of the protests, even though they inconvenience her. She is a Hongkonger, like the protesters from whom she seeks acceptance while acting on their behalf.

## 8. Discussion of the Results

Our purpose in this paper was to understand how Hong Kong's 2019 protest movement influenced the identities of a sample of minoritized youth. We set the paper within the framework of Lacan's identity theory in which it is assumed that identity is constructed largely by consciousness-raising processes under the influence of either major external forces (the "OTHER") or internal processes related to the discursive experiences of individuals (the "other"). We have not sought to make generalized statements about identities or identity formation. Rather, our focus has been on understanding identities in the contexts that formed them at a particular point in time. We now wish to highlight several key issues that help to inform our understanding of identities in the context of Hong Kong's protest movement.

### 8.1. Intragenerational Differences

Reporting on the attitudes of ethnic minority individuals and groups to the 2019 protests highlighted the intergenerational differences between youth and older members of the community [15].

What we suggest here, however, is that youth identities are not essentialized, but rather fragmented: intragenerational differences challenge the dichotomy of intergenerational differences. The sample of minoritized youth used in this study showed how individual identities were influenced in multiple ways by the protests, resulting in different actions, different attitudes, and different reflections.

One possible explanation for the narrative related to intergenerational differences is that at the beginning of the protests minority youth support was unanimous. This was not true for the older sample since many were opposed from the beginning. For some of our sample, major changes took place for some participants, especially where they thought their identity was being challenged. Again, this did not work for everyone in the same way. For at least one of our participants, there was an eventual outright rejection of the protesters, but for another, there was no questioning of her initial support. Another reported being "confused" by the violent elements in the protests since they challenged her preferred identity of living in a safe and secure Hong Kong. These were all changes over time for our participants, suggesting that intergenerational differences depended on the point of time in which they were observed or noted.

### 8.2. Constructions of Hong Kong as a Key Influence but Not Always on Identity

As the participants talked about their experiences and attitudes towards Hong Kong, they demonstrated a strong sense of belonging to the city. Sense of belonging has been defined as "emotional attachment, about feeling 'at home'" [16]. It has been pointed out that "belonging is always in relation to something outside the self (a place—in the social as well as geographical sense—and is therefore always 'located')" [17]. There are differences of opinion about whether this kind of belonging is the same as identity. Some have argued for a strong connection between belonging and identity [16], while others [17] have argued they are different. A Lacanian perspective would agree that there is a difference between belonging and identity largely because of the external nature of belonging. For Lacan, identity formation is an internal process, related to processes of consciousness and reflection.

All the cases considered in this study demonstrated a sense of belonging to Hong Kong. Yet there were differences in terms of identity formation. Malolos' case showed a deep and unquestioning commitment to the protest movement. Her identity was linked to her being accepted as a "Hongkonger". Yet for Amitra, the protests challenged her identity that was embedded in a safe and secure Hong Kong rather than what she perceived to be an

increasingly violent city under the influence of the protests. Aqidah also experienced this clash leading to a confused identity—she was supportive of the protesters but not of what they were doing to the city. Mizan asserted his Islamic identity alongside his engagement and support for the protesters. Thus sense of belonging emerged supporting the view that externalities and identity cannot always be equated.

### 9. Agency in Identity Formation

The cases in this study have shown that identities emerge in multiple and unpredictable ways. The process of the interviews allowed each interviewee to demonstrate how they thought about their experiences and with what effects. These discursive experiences affected individuals differently, as did the protest as a major social and political force. These individual reactions influenced identity in different ways. There is a degree of agency in these reactions in the sense that neither the protests nor the individual experiences were irresistible in their effects.

We would argue that such agency is enabled by the internal processes of individual reflection and consciousness raising. In the cases shown here, these processes led to positive outcomes in the sense of helping to affirm emerging identities. As such, they offer the possibility of resistance to external forces and even pressures. More needs to be known about these internal processes and how they can be developed. This represents an important agenda for future research relating to identity.

### 10. Implications

This study has reinforced the view articulated by Stuart Hall [10] that identity is contingent. Unlike citizenship, it is not a given category. Identities develop on account of circumstances and experience, but not simply because of them. Parents, educators, and policymakers need to appreciate this point when they seek to mold and shape young people to fit preconceived notions of who and what youth should be. Young people have the capacity to counter such pressures: this is a key learning from this study and one that needs to be taken aboard by the multiple groups that seek to influence them. Young people are not "blank slates": they will write on their own slates in their own way.

The theoretical framing of this study offers potential for further development in the specific context of research on identity. Understanding how internal processes of reflection work in relation to identity development can help to appreciate the ways in which individuals make decisions about their identities. Currently, we know that such decisions are made, but little is known about how they are made. Additional studies using the Lacanian framework that was the basis for this study may help to answer this question.

### 11. Limitations of the Study

The cases presented are not generalizable, either within the broader population from which the samples were drawn or the entire population of Hong Kong. The sample size is small and there is a lack of gender balance in the sample.

### 12. Future Research

The research reported here represents the beginning of a new direction for Hong Kong's minoritized population. In the future, more ethnic groups should be involved to understand more fully how different ethnicities influence identity formation. In addition, there should be a focus on the influence of gender on identity formation in the local context. Coupled with a focus on other factors, such as socioeconomic status, it would then be possible to explore issues of intersectionality. Working with researchers in other contexts, it is important to explore the impact of different contexts on the way ethnic minorities and their identities are formed.

## 13. Conclusions

Identity development in the context of Hong Kong's recent protest movement is an important social and political issue. We have shown here that its impact on the cases we presented is important, although different for each case. The protest movement in Hong Kong is the most significant political upheaval since the city's return to China. Yet engagement in and with the movement did not always win the hearts and minds of the participants in this study. They grappled with the issues in different ways thus showing how personal identity is not always subjugated to external pressures and in particular others' constructions of what is good for society. This study has shown that communities seeking to achieve common ends and shared goals should not underestimate the power of personal identities to create their own realities.

**Author Contributions:** Conceptualization, methodology and formal analysis, original draft preparation, K.J.K.; writing review and editing, J.C.G.; project administration, funding acquisition, M.K.B. All authors have read and agreed to the published version of the manuscript.

**Funding:** The research reported here was supported by the Public Policy Research Funding Scheme from the Policy Innovation and Co-ordination Office of the Government of the Hong Kong Special Administrative Region (Project Number: SR2020.A5.008). The views presented are those of the authors and not the funding body.

**Institutional Review Board Statement:** The study was conducted in accordance with the Declaration of Helsinki, and approved by the Human Ethics Research Committee of The Education University of Hong Kong, ethics approval reference Number 2019-2020-0206.

**Informed Consent Statement:** Informed consent was obtained from all subjects involved in the study.

**Data Availability Statement:** The data is the property of the Hong Kong SAR Government.

**Conflicts of Interest:** The authors declare no conflict of interest.

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
