# Peer review of "Identities in Troubled Times: Minoritized Youth in Hong Kong’s “Summer of Protest”"

_societies, doi:10.3390/soc13100217_

Round 1

Reviewer 1 Report

This article explores an important and relatively under-recognized area.

It is generally well organized and mostly quite well written. There are, however, some typographical errors, e.g., on line 54 I think “of just of” should read as “of just over” and the whole sentence in which that phrase appears is unclear. I am not sure it is reasonable to suggest that the 2019 protests were “a single event” (line 41). In line 111 I am not sure what is meant by “We four interviewees”. On line 173, I think “in the studies form” should read “in the studies from”. There is some vagueness (e.g., references to “other reports” and “the reports” lines 106-7 are unclear). I have not highlighted all the errors I have seen. I suggest that a thorough check would be helpful.

The characterization of minority could be clearer. Much is made of “a dominant Chinese citizenry” (line 73). I think this needs to be handled carefully. I think it would be fair to suggest that some make much of the distinction between the Cantonese speaking long established population and those who are relatively newly arrived from the Chinese mainland. In this article that focuses on the protests in 2019 a little more refinement and precision could be useful. This would also help in the clarification of - and justification for - the central purpose of this article. As it stands, things are stated rather than explained.

A huge amount of complex and contested material is summarized between lines 116 and 170. There is nothing here with which I would take strong exception but I feel that the author is risking unhelpfully reducing significant philosophical debates to rather simplistic positions. I fear that the data that has been collected will in such a brief article not be able to stand the pressure to achieve theoretical significance. Perhaps it would help to add a few words to acknowledge these complexities and, given the available space, the need to simplify matters.

I cannot understand why there is little or nothing about the methods used in this study. There is nothing about sampling, data collection, ethics, analysis, validity, reliability and so on.

 The thumbnail sketches of the people who supplied data together with quotations from their responses are very interesting. It is unfortunate that this material is not connected directly and meaningfully to the discussion of issues that takes place from line 316. I am not convinced that all of the comments made by the respondents and by the authors could reasonably be seen to be centrally about identity. Some of what is shown is simply what people think about the protests.

The discussion is limited. There is a lack of focus. For example, I am really not sure what is meant in this context by using the words the personal is political (line 408). Of course, that is true and we always need to be alert to the views of individuals, but what is the precise point being made in relation to identity formation in the context of the 2019 actions?

The article addresses a vitally important matter. We need more work on this topic. But we also need more depth and more precision. It is not helpful to read a few comments that are then inadequately linked to weighty theoretical matters.

I think there is a decision to be made about what could be done to the article. One approach would be for the author to undertake a fundamental review in which there is proper academic reflection. More realistically, I suggest a thorough proof read; explaining more clearly the purpose of the piece; providing a description and explanation of methods; providing, with appropriate modesty, references to theories of identity formation; highlighting (on the basis of data analysis) a few key areas that would warrant further research.

Thorough proof reading is necessary

Author Response

Comments to each comment is contained in the attachment.

Reviewer 2 Report

This is an article of paramount importance which contributes to research on identities in troubled times: minoritized youth in Hong Kong’s “Summer of Protest.” While this paper could be inspirational and empowering in terms of the impact of multiple identities and multiple belongings of minoritized youth in Hong Kong in troubled times, I suggest a minor revision for this manuscript. In the following, I am going to make one recommendation if the article is going to be revised and resubmitted. I will also make recommendations for the author/s for future inquiries and writing about inquiries.

First of all, I strongly suggest that the entire manuscript need very close editing for grammatical errors and other stylistic features. For example, nowadays, “identity” might need to be changed to “identities.” For another example, “their way” should be “their ways.” The manuscript could also be edited in-house. Above all, this is an extremely important and timely article which, I believe, contributes to research on multiple identities and multiple belongings of minoritized youth in troubling times not only in Hong Kong but also in other contexts in the world. I recommend a minor revision for the manuscript.

Now I am going to make recommendations for the author/s to think about as she/he/they engage in future inquiries and writing about inquiries. First of all, I would like to suggest that the author/s bring in more literature written by ingenious people in local contexts rather than those written mainly by outsiders such as Europeans [e.g., Asia as method (Chen, 2010), empire (Coloma, 2013), AsianCrit (Iftikar & Museus, 2018) as method, and decolonization and globalization (Lin & Martin, 2005)]. Secondly, there are abundant Asian scholars who write about the complexities of multiple identities and belongings rather than referring to the works of Lacan. Thirdly, the author/s might want to consider to write about “creative insubordination” or “the power of fictional narrative” in terms of inquiries into and writing about social movements in troubling times.

References

Chen, K. (2010). Asia as method: Toward deimperialization. Duke University Press.

Coloma, R. S. (2013). Empire: An analytical category for educational research. Educational Theory, 63(6), 639-657. https://doi.org/10.1111/edth.12046

Iftikar, J.S. & Museus, S. D. (2018) On the utility of Asian critical (AsianCrit) theory in the field of education, International Journal of Qualitative Studies in Education, 31(10), 935-949, DOI: 10.1080/09518398.2018.1522008. https://doi.org/10.1080/09518398.2018.1522008

Lin, A., & Martin, P. W. (2005). Decolonisation, globalisation: Language-in-education policy and practice. Multilingual Matters.

This is an article of paramount importance which contributes to research on identities in troubled times: minoritized youth in Hong Kong’s “Summer of Protest.” While this paper could be inspirational and empowering in terms of the impact of multiple identities and multiple belongings of minoritized youth in Hong Kong in troubled times, I suggest a minor revision for this manuscript. In the following, I am going to make one recommendation if the article is going to be revised and resubmitted. I will also make recommendations for the author/s for future inquiries and writing about inquiries.

First of all, I strongly suggest that the entire manuscript need very close editing for grammatical errors and other stylistic features. For example, nowadays, “identity” might need to be changed to “identities.” For another example, “their way” should be “their ways.” The manuscript could also be edited in-house. Above all, this is an extremely important and timely article which, I believe, contributes to research on multiple identities and multiple belongings of minoritized youth in troubling times not only in Hong Kong but also in other contexts in the world. I recommend a minor revision for the manuscript.

Now I am going to make recommendations for the author/s to think about as she/he/they engage in future inquiries and writing about inquiries. First of all, I would like to suggest that the author/s bring in more literature written by ingenious people in local contexts rather than those written mainly by outsiders such as Europeans [e.g., Asia as method (Chen, 2010), empire (Coloma, 2013), AsianCrit (Iftikar & Museus, 2018) as method, and decolonization and globalization (Lin & Martin, 2005)]. Secondly, there are abundant Asian scholars who write about the complexities of multiple identities and belongings rather than referring to the works of Lacan. Thirdly, the author/s might want to consider to write about “creative insubordination” or “the power of fictional narrative” in terms of inquiries into and writing about social movements in troubling times.

References

Chen, K. (2010). Asia as method: Toward deimperialization. Duke University Press.

Coloma, R. S. (2013). Empire: An analytical category for educational research. Educational Theory, 63(6), 639-657. https://doi.org/10.1111/edth.12046

Iftikar, J.S. & Museus, S. D. (2018) On the utility of Asian critical (AsianCrit) theory in the field of education, International Journal of Qualitative Studies in Education, 31(10), 935-949, DOI: 10.1080/09518398.2018.1522008. https://doi.org/10.1080/09518398.2018.1522008

Lin, A., & Martin, P. W. (2005). Decolonisation, globalisation: Language-in-education policy and practice. Multilingual Matters.

Author Response

All comments to both reviewers are in the attachment.

Round 2

Reviewer 1 Report

The revised version is much better than the original.

I wish to raise the following:

·         I hope that the additional details given for individuals does not lead to the possibility of identification by the authorities

·         I do not know why Lacan (and not as reviewer 2 suggests other scholars, including those who are more familiar with Asia, have not been used)

·         Now that the methods of the project are clearer, it is even more important to think about the relationship between the issues raised in the discussion section and the data on which those arguments are based. Specifically, for example, I am not convinced that we can place very much weight on claims regarding intra or intergenerational differences. Generally, the links between data and argument are not always persuasively demonstrated.

·         We have some very interesting comments about what a small group of people think about their place/role in Hong Kong in the context of protests. Whether that can realistically be said to constitute insights regarding identity is for me an open question.

·         I think the study takes a small group of people with very varied experiences and finds that they have a range of ideas/perceptions. I think it is reasonable for the authors to encourage readers to think about those things including differences (considering the role of age, agency and reaction to place in the development of them). That is a worthwhile thing to do. I am left a little under-whelmed by what the article reveals about identity.  

·         My advice would be for the authors to commit themselves a little more fully to the idea of an exploration of the views of a group with the intention of raising questions about identity.

Author Response

Responses were submitted as a file using the button below. 
